# Burden, risk factors, and clinical outcomes of pediatric malaria in Nigeria: A systematic review and meta-analysis protocol

Olayinka Rasheed Ibrahim[1]*, Jubril Abdulkareem[2], Amudalat Issa[3],
Michael Abel Alao[4]

1 Department of Pediatrics, Division of Clinical Medicine, University of Global Health Equity, Kigali, Rwanda, 2 Department of Medicine and Surgery, University of Ilorin, Ilorin, Nigeria, 3 Department of Pediatrics, Children Specialist Hospital, Ilorin, Nigeria, 4 Department of Pediatrics, University College Hospital, Ibadan, Nigeria

* ibroplus@gmail.com

## Abstract

### Background

Nigeria ranks number one globally in malaria burden, with the exact burden, especially for hospitalization, unknown. This systematic review and meta-analysis will pool laboratory-confirmed pediatric malaria data across Nigeria using standardized World Health Organization (WHO) case definitions and random-effects modeling to generate precise national and subgroup-specific estimates of prevalence, associated factors, and outcomes, thereby addressing a critical evidence gap.

### Methods and analysis

A comprehensive search strategy using MeSH terms, text words, and entry terms will be applied across six databases: PubMed, Embase, EBSCOhost, Google Scholar, Web of Science, and Scopus. Eligible studies will be observational and interventional studies, without language restriction, from inception through June 30, 2025. Non-English studies will be translated using professional translation tools [Google Translate (academic mode) and Microsoft Translator] and verified by native speakers. The primary outcome is the pooled prevalence of pediatric malaria [overall, severe malaria vs. uncomplicated]. Secondary outcomes include factors influencing clinical presentation and outcomes, as well as the effects of moderators such as age, sex, socioeconomic status, and geographic location. Data extraction will capture study characteristics, participant demographics, and outcome measures. Methodological, clinical, and statistical heterogeneity will be assessed. Risk of bias will be assessed using the Newcastle-Ottawa Scale (NOS) adapted for observational studies, and the Cochrane Risk of Bias tool (RoB 2.0) for interventional studies. Publication bias will be examined using funnel plots and Egger's regression. Pooled estimates will

**Data availability statement:** No datasets were generated or analysed during the current study. All relevant data from this study will be made available upon study completion.

**Funding:** The author(s) received no specific funding for this work.

**Competing interests:** The authors have declared that no competing interests exist.

**Abbreviations:** PRISMA-P: Preferred Reporting Items for Systematic Review and Meta-Analysis Protocols), NOS: Newcastle–Ottawa Scale, PROSPERO: International Prospective Register of Systematic Reviews.

be reported with 95% confidence intervals. This will summarize the data on pediatric malaria in Nigeria. Using the random-effects models, the pooled prevalence, along with the 95% CI and the $I^2$ for the test of heterogeneity, will be reported. We will report the meta-regression analysis of factors influencing pediatric malaria in Nigeria.

## Conclusions

This study will provide robust data on pediatric malaria and associated factors in Nigeria. The findings from this study will inform the country's policy and public health approach as the nation strives to eliminate malaria in line with the WHO's goals. PROSPERO Registration Number: CRD420251163322.

---

## Introduction

Malaria is a parasitic infection caused by *Plasmodium spp.* and remains a public health concern in the tropics. The 2024 World Health Organization (WHO) malaria report estimated 263 million (95% confidence interval: 238 million to 294 million) malaria cases and 597,000 related deaths in 2023.[1] The significant burden of malaria disproportionately affects sub-Saharan Africa, with Nigeria ranked as number one and accounting for 27% of the global cases.[2] In 2023, the country reported an estimated 68.1 million (95% confidence interval: 49.1 million to 92.6 million) malaria cases and approximately 185,000 deaths.[1] In Nigeria, like the rest of the world, the significant burden of malaria affects the pediatric age group, where it is as high as 80% [2].

Despite the available estimates on malaria in Nigeria, the data do not delineate to reflect the actual burden of pediatric severe malaria at the hospital levels, uncomplicated malaria, associated factors, various complications, and clinical outcomes. [1, 2] Available studies conducted among children and adolescents in Nigeria have demonstrated significant variations in the local burden, complications, and outcome of pediatric malaria across the six geopolitical zones in Nigeria and within the same zone.[2–5] With variations in the burden reported across the country, the exact burden of severe malaria, uncomplicated malaria, associated factors, complications, and clinical outcomes in the country remains largely unknown, which hampers resource allocation and policy formation.

This observation, therefore, highlighted the need to synthesize data from studies of the Nigerian pediatric population with malaria and its associated factors. The synthesis of this data will also enable the examination of trends over the years, assess the country's progress, inform policy interventions, and ultimately ensure that the country achieves its goal of eliminating malaria. We therefore raised the following research questions: What is the pooled prevalence of pediatric malaria in Nigeria [severe and uncomplicated]? What are the associated factors for pediatric malaria in Nigeria? What is the trend of pediatric malaria in Nigeria [severe and uncomplicated]? What are the outcomes [discharged, deaths, and complications, e.g., neurological] of pediatric severe malaria in Nigeria?

This study intends to conduct a systematic review and meta-analysis of laboratory-confirmed pediatric malaria data (severe and uncomplicated) across Nigeria using standardized World Health Organization (WHO) case definitions and random-effects modeling to generate precise national and subgroup-specific estimates of prevalence, associated factors, and outcomes, thereby addressing a critical evidence gap.

## Methods

The review was registered with the International Prospective Register of Systematic Reviews (PROSPERO: CRD420251163322) to enhance methodological transparency and prevent duplicate research. Any protocol amendments will be documented and updated accordingly in the registry.

### Eligibility criteria

The PICOS will be used to define the eligibility criteria for this study, which will include:

Population (P)—Published studies conducted in Nigeria that include children and adolescents aged 0–18 years diagnosed with malaria (severe and uncomplicated malaria).

Intervention (I)—Not applicable based on the topic outline. However, socioeconomic factors, age, sex, geopolitical zones, and diagnostic methods will be examined as potential confounders.

Comparator (C)—Comparisons across sex, age groups, socioeconomic class, geopolitical zones, diagnostic methods, and Plasmodium species.

Outcomes (O)—Primary outcome will include

1. The pooled prevalence of pediatric malaria [overall, severe malaria vs. uncomplicated].

The secondary outcomes

1. The associated factors [sociodemographic—age, sex, socioeconomic status, and educational level; environmental/climatic factors—dry and rainfall seasons; behavioral-related factor—insecticide-treated net use] for pediatric malaria

2. Outcomes of severe malaria, including discharge, mortality, and complications. To attribute complications, discharge, or death to malaria, we will restrict outcome analyses to those who reported malaria as the primary diagnosis.

3. The trend of pediatric malaria over time

4. Subgroup analysis of prevalence by age group, sex, socioeconomic status, seasonality (rainfall vs. dry season), geopolitical zones, diagnostic method, and co-morbidity context.

Study design (S): This review will include quantitative observational and interventional studies, including cross-sectional, cohort, case-control, and longitudinal studies, conducted in Nigeria and published in any language. This review will primarily include observational and interventional studies that report extractable epidemiological data.

Non-English studies will be translated using professional translation tools, Google Translate (academic mode), and Microsoft Translator. Native-language speakers will verify the translated studies to ensure accuracy during data extraction.

### Exclusion criteria

The following studies will be excluded from this study.

1. Case reports, case series, editorials, commentaries, qualitative studies, and review articles (including narrative and systematic reviews)

2. Studies without clearly defined population denominators

---

3. Studies that are irretrievable after reasonable attempts to contact the corresponding authors

4. Studies with a sample size of fewer than 100 participants. This will be excluded, as very small studies may be more prone to selection bias, overestimation of effect sizes, and publication bias.[6] Excluding studies with fewer than 100 participants will enhances the methodological robustness of the meta-analysis and aligns with practices commonly adopted in epidemiological prevalence reviews conducted in high-burden infectious diseases.[6]

5. Studies without laboratory confirmation of malaria [microscopy, rapid diagnostic tests for malaria, or polymerase chain reaction]

6. Studies on asymptomatic malaria [defined as the presence of malaria parasites in an individual's blood without symptoms or signs].[7] This population will be excluded as asymptomatic malaria studies represent a distinct epidemiological entity without clinical manifestations or outcomes, and their inclusion would inflate prevalence estimates and introduce heterogeneity, thereby limiting accurate assessment of the clinical burden and outcomes of pediatric malaria.

## Information sources

A comprehensive literature search will be conducted across the following databases: PubMed, Embase, EBSCOhost, Google Scholar, Web of Science, and Scopus. In addition, manual searches will be performed by screening the reference lists of included studies and relevant reviews to identify any additional eligible articles and grey literature sources.

## Search strategy

The search strategy will combine Medical Subject Headings [MeSH] terms, keywords, and entry terms adapted for each database to ensure sensitivity and specificity. The terms and keywords will include variations and combinations of the following: Geographic terms: "Nigeria"; Population terms: "newborns," "infants," "children," "adolescents"; Disease terms: "malaria," "severe malaria," "uncomplicated malaria"; Outcome and associated factor terms: "sex," "socioeconomic status," "geopolitical zones," "associated factors," "complications," "outcomes," "discharge," "deaths," and "hospitalization."

An example of the PubMed search string (to be adapted for other databases) is as follows: ("Nigeria"[Mesh] OR "Nigeria" OR "Nigerian") AND ("Child"[Mesh] OR "Infant"[Mesh] OR "Adolescent"[Mesh] OR "Pediatrics"[Mesh] OR "Infant, Newborn"[Mesh] OR child*[tiab] OR infant*[tiab] OR adolescent*[tiab] OR pediatric*[tiab] OR paediatric*[tiab] OR newborn*[tiab] OR neonate*[tiab] OR "school-age"[tiab] OR "under-five"[tiab] OR "under five"[tiab] OR "young children"[tiab]) AND ("Malaria"[Mesh] OR "Malaria, Falciparum"[Mesh] OR "Malaria, Vivax"[Mesh] OR "Plasmodium falciparum"[Mesh] OR malaria[tiab] OR "severe malaria"[tiab] OR "uncomplicated malaria"[tiab] OR "Plasmodium falciparum"[tiab] OR "Plasmodium infection*"[tiab] OR "malaria infection"[tiab] OR "malaria burden"[tiab] OR "malaria morbidity"[tiab] OR "malaria mortality"[tiab] OR "malaria hospitalization"[tiab]) AND ("Risk Factors"[Mesh] OR "Socioeconomic Factors"[Mesh] OR "Sex Factors"[Mesh] OR "Epidemiology"[Mesh] OR "Disease Susceptibility"[Mesh] OR "Social Determinants of Health"[Mesh] OR determinant*[tiab] OR "associated factor*"[tiab] OR "predictor*"[tiab] OR "correlate*"[tiab] OR "socioeconomic status"[tiab] OR "education level"[tiab] OR "household income"[tiab] OR "parental occupation"[tiab] OR "sex difference*"[tiab] OR "gender difference*"[tiab] OR "clinical outcome*"[tiab] OR "treatment outcome*"[tiab] OR mortality[tiab] OR death*[tiab] OR fatal*[tiab] OR discharge[tiab] OR complication*[tiab] OR "neurological sequelae"[tiab] OR hospitalization[tiab] OR "disease severity"[tiab] OR "case fatality"[tiab]) AND ("cross-sectional studies"[Mesh] OR "cohort studies"[Mesh] OR "case-control studies"[Mesh] OR "observational study"[Publication Type] OR "interventional studies"[Mesh] OR "clinical trials "[Mesh] OR "longitudinal studies"[Mesh] OR "epidemiologic studies"[Mesh] OR "survey"[tiab] OR "prevalence"[tiab] OR "incidence"[tiab] OR "retrospective"[tiab] OR "prospective"[tiab]) –(Appendix 1)

There will be no language restrictions during the literature search. All relevant studies, irrespective of language, will be included. Non-English studies identified during the screening will be translated into English before data extraction to minimize selection and translation bias.

## Study Selection

Following the literature search, all retrieved records will be exported to the Rayyan software for systematic screening and management. The screening will be done using different levels to enhance transparency as follows: Level 1 would involve screening of identified studies for the study design inclusion (quantitative observational and interventional studies); Level 2 will involve screening of identified studies in the titles and abstracts using entry terms, keywords, and MeSH terms; Level 3 will involve further screening of the contents of articles by reading the full article using the same search strategy; Level 4 will involve snowballing of literature on references from included studies; Level 5: Studies will be screened at outcome levels to select those that reported the primary outcome with or without secondary outcomes; Level 6 will involve grey literature that reports the primary outcome and/or secondary outcomes. The screening process will be conducted independently by two reviewers (AI and JA) to identify studies that meet the inclusion criteria. A third reviewer (ORI) will resolve any discrepancies or disagreements between the two reviewers. The study selection process will adhere to the Preferred Reporting Items for Systematic Reviews and Meta-Analyses (PRISMA 2020) guidelines.

## Data collection process

Data extraction will be carried out independently by two reviewers [AI and JA] using a predefined data extraction form that will be created in Microsoft Excel. The data will include the first author's name, year of publication, total number of study participants, number of cases [severe/uncomplicated malaria], study settings, study design, diagnostic methods for malaria, age, sex, clinical outcomes (e.g., discharge, death, neurological sequelae), states/geopolitical zones of the studies, and associated factors (demographics, socioeconomic status, environmental and climatic factors (rainfall pattern/seasonality, rural vs. urban), ownership of bed nets, utilization of bed nets, and parasite species). Any discrepancies in the extracted data will be resolved in consultation with the fourth reviewer (MAA).

## Operational definition

The definitions of asymptomatic, uncomplicated malaria and severe malaria will follow the WHO definitions.[8]

Asymptomatic malaria will be defined as the presence of malaria parasites in an individual's blood without symptoms and signs.[7]

Uncomplicated malaria will be defined as the presence of malaria symptoms and confirmed malaria parasites, without signs of vital organ dysfunction. [8]

Severe malaria will be defined as the presence of *Plasmodium falciparum* parasitemia with vital organ dysfunctions, as indicated in the WHO guidelines for severe malaria. [8]

## Risk of bias assessment and methodological quality assessment

The methodological quality and risk of bias of the included studies will be independently assessed by two reviewers [AI and JA] using the Newcastle-Ottawa Scale (NOS) adapted for observational studies, while the Cochrane Risk of Bias tool (RoB 2.0) will be used for interventional studies.[9] The NOS tool evaluates observational studies based on three key domains: selection of study groups, comparability, and ascertainment of outcomes or exposures.[9] Each study will be assigned a score based on the NOS criteria, and the results will be categorized as low, moderate, or high risk of bias. Inter-rater agreement between the two reviewers will be assessed using Cohen's kappa statistic to evaluate the consistency of screening and risk-of-bias assessment. Where significant differences exist between the two rates, the third reviewer (ORI) will serve as an arbiter, and their final rating will be used for the study.

## Outcome definitions and attribution

Clinical outcomes will include discharge, mortality, and complications. Only outcomes reported among laboratory-confirmed malaria cases (diagnosed by microscopy, rapid diagnostic test [RDT], or polymerase chain reaction [PCR]) will be eligible for inclusion in the outcome synthesis. For severe malaria outcomes, studies will be included if they meet the WHO severe malaria criteria and report malaria as the primary diagnosis.

## Outcomes will be categorized based on timing

*In-hospital outcomes:* discharge alive, in-hospital death, and complications occurring during the index malaria admission (primary time frame). If outcome timing is not explicitly stated but clearly refers to events "during admission," it will be classified as in-hospital. Outcomes with unclear timing will be flagged and assessed in sensitivity analyses.
*Handling Co-morbidities and Mixed-Cause Deaths:* Where reported, significant co-morbidities (e.g., bacteremia/sepsis, pneumonia, malnutrition, HIV, sickle cell disease) will be extracted. If studies provide stratified outcomes for malaria-only versus malaria with co-morbidities, outcomes will be preferentially extracted. When stratification is unavailable, studies will be extracted and captured as "all-cause mortality."

Deaths will be classified as malaria-specific mortality and all-cause mortality.

## Data synthesis and statistical analysis

Data extracted from the predefined Excel spreadsheet will be imported into R for analysis using the "meta" and "metafor" packages.[10] The pooled prevalence of pediatric malaria (overall, severe, and uncomplicated) will be estimated, along with 95% confidence intervals (CIs). The odds ratio (OR), along with 95% CI, will be used to estimate the association between factors and malaria. If significant heterogeneity is observed among the included studies, a random-effects model (DerSimonian–Laird method) will be used to compute pooled prevalence estimates, providing a more conservative estimate of variability.[11] Forest plots will be generated to visually display the pooled estimates, corresponding confidence intervals, and prediction intervals.

## Assessment of heterogeneity, handling, and effect measures

Statistical heterogeneity will be assessed using $I^2$, $\tau^2$, Cochran's Q, and prediction intervals.[12] An $I^2$ value greater than 50% will be considered indicative of substantial heterogeneity.[13] Substantial heterogeneity is anticipated due to variation across Nigerian geopolitical zones, diagnostic methods, healthcare settings, and time periods. Where substantial heterogeneity is observed, subgroup analyses and mixed-effects meta-regression will be conducted to explore potential sources of variation, including age group, sex, socioeconomic status, geopolitical zone, diagnostic methods, and study design. Meta-analysis will not be performed when clinical or methodological heterogeneity precludes comparability (e.g., incompatible outcome definitions or timeframes), or when fewer than three studies contribute comparable data. In such cases, findings will be synthesized narratively and summarized in structured tables.

For associated factors, effect estimates will be pooled by measure type (odds ratios, risk ratios, hazard ratios, or prevalence ratios) using random-effects models. Adjusted estimates will be pooled separately from unadjusted estimates. Where appropriate, effect measures will be converted to a common metric to facilitate pooling, provided sufficient data are available and the assumptions underlying the conversion are reasonable (e.g., low outcome prevalence for odds ratio–to–risk ratio conversion).[14] Converted and non-converted estimates will be examined in sensitivity analyses. When pooling is not appropriate—for example, due to substantial heterogeneity in effect measures with insufficient studies per measure, or non-comparable exposure or outcome definitions—a narrative synthesis will be undertaken.[14] Effect estimates will be summarized in structured tables, including the direction and magnitude of associations and the variables included in adjustment models.

## Assessment of publication bias and sensitivity analysis

Publication bias will be evaluated visually using funnel plots and Egger's test.[15] To assess the potential influence of missing studies, a trim-and-fill analysis will be performed.[16] Sensitivity analyses will be conducted to evaluate the impact of studies with a higher risk of bias on the pooled estimates.[17] Additionally, a leave-one-out sensitivity analysis will be performed to assess the influence of each study on the overall pooled estimate and to evaluate the robustness of the findings. To assess the potential impact of excluding smaller studies, a sensitivity analysis will be conducted by re-including studies with sample sizes fewer than 100 participants that otherwise meet all eligibility criteria (including laboratory confirmation of malaria). The sensitivity analysis will compare pooled prevalence and outcome estimates with and without small studies. This will assess whether inclusion of small studies materially alters the direction or magnitude of the findings. Results from this sensitivity analysis will be reported alongside the primary analysis to evaluate the robustness of the conclusions. If substantial differences are observed, these will be discussed as potential evidence of small-study effects and interpreted with caution.

## Subgroup and meta-regression analyses

Prespecified subgroup analyses will be conducted, where data permit, to explore heterogeneity in the pooled prevalence and outcomes of pediatric malaria.[18] Subgroups will include age group (neonates, infants, under-five children, school-age children, adolescents), sex, geopolitical zone, study design, seasonality (rainfall vs. dry season), diagnostic method (microscopy, RDT, PCR), malaria species, risk-of-bias category, and year of study. Subgroup meta-analyses will be performed using random-effects models and only when at least three studies contribute data to a given subgroup; otherwise, findings will be summarized narratively.

To further investigate sources of heterogeneity, mixed-effects meta-regression analyses will be conducted using study-level covariates.[19] Prespecified covariates will include age group, sex, year of study, geopolitical zone, diagnostic method, study design, and seasonality. Univariable meta-regression will be performed when at least 10 studies report the covariate of interest. Multivariable meta-regression will be conducted only when sufficient data are available (a minimum of approximately 10 studies per covariate) to avoid model overfitting. All subgroup and meta-regression analyses will be considered exploratory and interpreted cautiously.

All statistical tests will be two-tailed, and a p-value $< 0.05$ will be considered statistically significant for all analyses.

## Status and timeline of the study

This protocol has just been registered in PROSPERO, with a preliminary title search to avoid duplication of previous work. Afterward, the plan is to publish the protocol, conduct a literature search (four weeks), record screening (four weeks), have a data abstraction period (four weeks), and analyze the results and share the findings of the study (four weeks) [Table 1].

## Ethical consideration and declaration

As this study involves the review and synthesis of data from previously published studies, ethical approval is not required. No primary data will be collected, and the review will not involve any direct interaction with human participants or access to confidential patient information.

## Results

The results of this systematic review and meta-analysis will be reported in accordance with the Preferred Reporting Items for Systematic Reviews and Meta-Analyses (PRISMA) 2020 guidelines. The findings from this study will be a detailed description of the literature search process, the number of records retrieved from each database, the total number of

**Table 1. Proposed timeline for the systematic review and meta-analysis.**

| Month | September 2025 | October 2025 | November 2025 | December 2025 | January 2026 | February 2026 |
|---|---|---|---|---|---|---|
| | Title search and registration (done) | | | | | |
| | | Submission for peer review (done) | | | | |
| | | | Literature search (yet to be done) | | | |
| | | | | Data abstraction (yet to be done) | | |
| | | | | | Data analysis (yet to be done) | |
| | | | | | | Submission of findings for publication (yet to be done) |

studies screened, and the number of studies included in the final analysis. A table summarizing the characteristics of the final studies included in the analysis will capture the number of participants, study design, malaria classification, geographic distribution, associated factors, diagnostic methods, complications, clinical outcomes (e.g., discharge, death, neurological sequelae), and risk of bias for each study.

For data synthesis, a random-effects model will be used to estimate pooled prevalence and 95% confidence intervals (CIs) for overall malaria, severe malaria, uncomplicated malaria, complications of severe malaria, associated factors, and clinical outcomes.

Subgroup analysis will include burden by age group, sex, socioeconomic status, geographic location, seasonality, use of insecticide-treated nets, study setting, study design, diagnosis, and year of publication.

A meta-regression analysis will be employed to assess the influence of these subgroup variables and other study-level covariates on the pooled prevalence estimates, thereby identifying factors that may contribute to heterogeneity across studies.

The findings will be presented as figures (forest plots, funnel plots, and fill-and-trim plots) and summary tables.

## Strengths and limitations

This systematic review and meta-analysis has some strengths. First, it will provide a comprehensive national synthesis of the burden, trends, associated factors, and outcomes of pediatric malaria in Nigeria, using a rigorous, transparent methodology aligned with the PRISMA 2020 guidelines. Second, the use of laboratory-confirmed malaria cases, strict outcome attribution criteria, and WHO-defined severity classifications will enhance the validity of prevalence and outcome estimates. Third, the application of random-effects models, prespecified subgroup analyses, and meta-regression will allow for exploration of heterogeneity across demographic, geographic, clinical, and methodological factors. Finally, multiple database searches and inclusion of grey literature will help minimize publication bias.

Despite the strength of this study, some limitations are anticipated. First, heterogeneity in study designs, diagnostic methods, and outcome definitions may contribute to substantial statistical heterogeneity, potentially limiting comparability across studies. Second, many included studies are expected to be hospital-based, which may overestimate malaria severity and mortality compared with community-based populations, thereby limiting generalizability. Third, variations in reporting of co-morbidities, follow-up periods, and outcome attribution may restrict pooling of some outcomes. Finally, temporal and regional data gaps may limit the ability to fully characterize trends in pediatric malaria across all geopolitical zones of Nigeria.

Despite these limitations, the planned review is expected to generate robust, policy-relevant evidence to inform malaria control and elimination strategies in Nigeria.

## Discussion

The findings from this systematic review and meta-analysis will provide a comprehensive, evidence-based estimate of the burden of pediatric malaria in Nigeria, encompassing both severe and uncomplicated cases. By synthesizing data from multiple studies across diverse settings and populations, this review will fill a critical knowledge gap in the epidemiology of malaria in pediatric age groups in Nigeria.

The results will be contextualized and compared with existing regional and global malaria data, highlighting similarities and differences in prevalence, associated factors, and outcomes. Potential explanations for observed variations—such as differences in diagnostic methods, transmission intensity, socioeconomic factors, and healthcare access—will be explored to offer an in-depth summary of malaria patterns in Nigeria's pediatric population.

The findings from the meta-regression analysis are expected to identify and model key determinants of malaria prevalence and severity, such as age, sex, socioeconomic status, geographic zone, and diagnostic method. These insights will be invaluable in informing policy formulation and revision, particularly as Nigeria continues to align its malaria control and elimination strategies with the WHO's global malaria targets.

We will also discuss in detail the research, policy, and public health implications of this study. Specifically, the study is expected to guide resource allocation, targeted interventions, and surveillance strategies for malaria prevention and management in children.

Finally, the discussion will also address the limitations of the review, including potential publication bias, heterogeneity in study methodologies, and regional data gaps, to ensure a transparent interpretation of the findings.

## Dissemination

The findings from this systematic review and meta-analysis will be disseminated widely to maximize their impact. Results will be submitted for publication in a peer-reviewed scientific journal and presented at national and international conferences related to infectious diseases, pediatrics, and public health. In addition, summaries of the findings may be shared with relevant stakeholders, including policymakers, malaria control programs, and public health institutions, to support evidence-based decision-making and malaria elimination efforts in Nigeria. The published article and its supplementary materials will make all data supporting the findings of this review available, ensuring transparency and reproducibility.

## Supporting information

**S1 File. PRISMA-P checklist uploaded.**
(DOCX)

**S2 File. Search strategy uploaded. a. The review work is self-funded. b. Sponsor: All the authors contributed to sponsoring the project. c. Guarantor of the review: ORI.**
(DOCX)

## Acknowledgments

We will acknowledge any individual that helps in this work but does not fulfill authorship criteria as listed by the International Committee of Medical Journal Editors.

## Author contributions

**Conceptualization:** Ibrahim Rasheed Olayinka, Amudalat Issa, Michael Abel Alao.

**Data curation:** Ibrahim Rasheed Olayinka, Jubril Abdulkareem, Amudalat Issa, Michael Abel Alao.

**Formal analysis:** Ibrahim Rasheed Olayinka, Jubril Abdulkareem, Amudalat Issa, Michael Abel Alao.

**Investigation:** Amudalat Issa.

**Methodology:** Ibrahim Rasheed Olayinka, Jubril Abdulkareem, Amudalat Issa, Michael Abel Alao.

**Project administration:** Ibrahim Rasheed Olayinka.

**Resources:** Ibrahim Rasheed Olayinka, Jubril Abdulkareem, Amudalat Issa, Michael Abel Alao.

**Supervision:** Ibrahim Rasheed Olayinka, Michael Abel Alao.

**Validation:** Ibrahim Rasheed Olayinka, Jubril Abdulkareem, Amudalat Issa, Michael Abel Alao.

**Visualization:** Ibrahim Rasheed Olayinka, Jubril Abdulkareem, Amudalat Issa, Michael Abel Alao.

**Writing – original draft:** Ibrahim Rasheed Olayinka, Jubril Abdulkareem, Amudalat Issa, Michael Abel Alao.

**Writing – review & editing:** Ibrahim Rasheed Olayinka, Jubril Abdulkareem, Amudalat Issa, Michael Abel Alao.

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
