## [Decision Letter · Decision Letter 0]

19 Dec 2025

Dear Dr. Olayinka,

We look forward to receiving your revised manuscript.

Kind regards,

Taiwo Opeyemi Aremu, MD, MPH, PhD

Academic Editor

PLOS One

Journal Requirements:

3. Please include captions for your Supporting Information files at the end of your manuscript, and update any in-text citations to match accordingly. Please see our Supporting Information guidelines for more information: http://journals.plos.org/plosone/s/supporting-information .

Additional Editor Comments:

Please make the description of eligible study designs fully consistent across the manuscript, reconciling the abstract (observational only) with the Methods section (observational and interventional).Make the description of language restrictions fully consistent across all sections, ensuring that the abstract reflects the policy of including studies "in any language" and briefly describing how non‑English articles will be translated and checked.Provide a clear operational definition of "asymptomatic malaria" and ensure this definition underpins its use as an exclusion criterion.Clarify and standardize what you mean by "associated factors," explicitly listing the prespecified factors (e.g. demographic, socioeconomic, environmental, clinical) and explaining how these will be analyzed (e.g. as covariates in subgroup or meta‑regression analyses).Clarify for outcomes such as discharge, mortality, and complications how you will ensure that these outcomes are attributable to malaria (e.g. restriction to lab‑confirmed clinical malaria, handling of co‑morbidities and mixed‑cause deaths) and define any time frames used.Justify the choice of excluding studies with fewer than 100 participants and, if feasible, pre‑specify a sensitivity analysis including these smaller studies to assess their impact.Make explicit in the Methods that full search strings for all databases are provided in a supplementary file (e.g. "Search string for the study‑malaria") and ensure that the example PubMed strategy in the text matches the supplementary material.Please strengthen the Data Synthesis and Statistical Analysis section by adding appropriate methodological references for the random‑effects model, the use of R ("meta" and "metafor" packages), and key measures such as I² and prediction intervals.Add supporting methodological references for the Assessment of Publication Bias and Sensitivity Analysis (e.g. Egger’s test, funnel plots, trim‑and‑fill, leave‑one‑out analyses) and briefly clarify how results from these procedures will be interpreted.Provide a more detailed, prespecified plan for subgroup analyses and any meta‑regression (e.g. listing subgroup variables, minimum number of studies per subgroup/covariate, and the basic modelling approach).Consider adding an explicit "Strengths and limitations" subsection that also summarizes the main anticipated limitations of the planned review (e.g. heterogeneity of definitions, risk of publication/language bias, hospital‑based data).Please revise the manuscript carefully for formatting, structure, and English language clarity (e.g. consistent terminology, numbering, headings) so that it meets PLOS ONE's readability and presentation standards.

Reviewer's Responses to Questions

**Comments to the Author**

1. Does the manuscript provide a valid rationale for the proposed study, with clearly identified and justified research questions?

Reviewer #1: Partly

Reviewer #2: Partly

Reviewer #3: Yes

2. Is the protocol technically sound and planned in a manner that will lead to a meaningful outcome and allow testing the stated hypotheses?

Reviewer #1: Partly

Reviewer #2: Yes

Reviewer #3: Yes

3. Is the methodology feasible and described in sufficient detail to allow the work to be replicable?

Reviewer #1: No

Reviewer #2: No

Reviewer #3: Yes

4. Have the authors described where all data underlying the findings will be made available when the study is complete?

Reviewer #1: Yes

Reviewer #2: Yes

Reviewer #3: Yes

5. Is the manuscript presented in an intelligible fashion and written in standard English?

Reviewer #1: No

Reviewer #2: Yes

Reviewer #3: Yes

You may also provide optional suggestions and comments to authors that they might find helpful in planning their study.

Reviewer #1: I noticed a couple of inconsistencies that should be clarified. For example, the abstract says only observational studies will be eligible, but the eligibility criteria section later mentions both observational and interventional studies. Similarly, the abstract limits inclusion to studies published in English, while the eligibility criteria section states that studies in any language will be included.

Some points also need more detail. The cutoff of 100 participants for exclusion is mentioned, but the rationale for this threshold isn’t explained. The secondary outcomes are also a bit vague. For instance, “associated factors” isn’t clearly defined, and for outcomes like discharge, mortality, and complications, it’s not clear how the authors will ensure these outcomes are directly attributable to malaria.

In the exclusion criteria, “asymptomatic malaria” should be explicitly defined to avoid ambiguity.

For the search strategy, I’d suggest adding a supplementary file that lists all search strings used across databases. That would make the process more transparent and reproducible for readers.

In the methods section, some areas like Data Synthesis and Statistical Analysis and Assessment of Publication Bias and Sensitivity Analysis don’t have any supporting references. Even if these are common methods, citing references would strengthen the protocol. I also don’t see where the subgroup analysis methodology is clearly and detailedly stated.

An “Expected Limitations” section would help.

Finally, there are many formatting and arrangement issues throughout the manuscript that don’t meet standards, which may lead to difficulty in reading for readers.

Reviewer #2: Dear Authors,

Well done for the effort in drafting this protocol.

Please address the concerns below.

Abstract: Line 26-28

Clarify how your study design will fill the gap (i.e. unknown exact burden).

Methods:

Line 32 reads "Eligible studies will be observational and published in English from inception till June 30, 2025."

Lines 105-107 also reads "Study design (S): This review will include quantitative observational and interventional studies, including cross-sectional, cohort, case-control, and longitudinal studies, conducted in Nigeria and

published in any language."

First statement has observational studies (no indication of interventional studies) and language restriction. The second statement has interventional studies and no language restriction.

Reconcile the two statements and include the necessary justifications.

Exclusion criteria:

What happens to studies involving adult malaria in your systematic search?

Line 116 reads "4. Studies with a sample size of fewer than 100 participants."

Justify clearly (with reference, if any) why you will exclude studies with specifically less than 100 sample size.

Line 119 reads "6. Studies on asymptomatic malaria."

What do you mean by asymptomatic malaria? Justify its exclusion.

Provide clarity in the two sentences below.

144 ... "Level 1 would involve screening of identified studies for the

145 study design quantitative observational and interventional studies would be accepted."

151 ... "The screened will be done

152 independently by two reviewers..."

In lines 176-177, citation is required for the statement "The NOS tool evaluates observational studies based on three key domains: selection of study groups, comparability, and ascertainment of outcomes or exposures."

All statements under authors’ contributions (lines 271-278) appear to suggest that search, data extraction, and analysis have already been carried on. Please rectify.

Reviewer #3: Thanks for your impactful study. Comments:

The abstract initially states inclusion of English-language studies only, while later sections state “any language” with translation plans.

Please ensure full consistency across the abstract, methods, eligibility criteria, and search strategy. The revised “no language restriction” approach is methodologically appropriate, but it must be reflected uniformly throughout the manuscript.

Given the wide variation expected across Nigerian geopolitical zones, diagnostic methods, healthcare levels, and time periods, heterogeneity is likely to be substantial.

Please predefine thresholds for when meta-analysis will not be performed due to excessive heterogeneity and clarify how such situations will be handled.

The protocol states that odds ratios will be pooled for associated factors. However, observational studies may report heterogeneous effect measures (OR, RR, HR).

Please clarify plans for handling different effect sizes (e.g., conversion methods, separate meta-analyses, or narrative synthesis).

The protocol addresses an important gap and has the potential to produce valuable evidence for malaria control policy in Nigeria.

**Do you want your identity to be public for this peer review?** For information about this choice, including consent withdrawal, please see our Privacy Policy

Reviewer #1: No

Reviewer #2: No

Reviewer #3: No

---

## [Author Response · Author response to Decision Letter 1]

2 Feb 2026

The Academic Editor

PLOS ONE

Dear Sir/Ma,

Re: PLOS ONE Decision: Revision required [PONE-D-25-56196]-[EMID:f1d693b70f52b0d9]

We are grateful for the detailed constructive feedback and criticism, which improved our protocol. We have taken into consideration all the editorial comments and reviewers’ comments and have addressed them in the revised copy,

Below are responses including where changes are made to the manuscript

Authors’ response: The manuscript has been formatted to be consistent with PLOS ONE’s style requirements.

Authors’ response

The Data Availability Statement has been provided in the submission form.

Authors’ response

All supporting documents have captured in the text and appropriate supplementary files uploaded

4. If the reviewer comments include a recommendation to cite specific previously published works, please review and evaluate these publications to determine whether they are relevant and should be cited. There is no requirement to cite these works unless the editor has indicated otherwise

Authors’ response

Many thanks-the reviewer comments did not include a recommendation to cite a specific previously published work.

Additional Editor Comments:

Please make the description of eligible study designs fully consistent across the manuscript, reconciling the abstract (observational only) with the Methods section (observational and interventional).

Authors’ response

We appreciate this observation with thanks. The manuscript has been revised, and consistency has been ensured in the eligible studies (lines 34-37)

Make the description of language restrictions fully consistent across all sections, ensuring that the abstract reflects the policy of including studies "in any language" and briefly describing how non-English articles will be translated and checked.

Authors’ response

We appreciate this observation with thanks. The search was without language restrictions, and this has been updated throughout the manuscript. Details of how non-English articles will be translated and checked have been added to the protocol (lines 34-37)

Provide a clear operational definition of "asymptomatic malaria" and ensure this definition underpins its use as an exclusion criterion.

Authors’ response

We have added a clear operational definition of asymptomatic malaria, which will be used for exclusion during screening (Line 136-141).

Clarify and standardize what you mean by "associated factors," explicitly listing the prespecified factors (e.g., demographic, socioeconomic, environmental, clinical) and explaining how these will be analyzed (e.g., as covariates in subgroup or meta-regression analyses).

Authors’ response

Thank you for the clarification. We have provided the details of the associated factors and further provided information on how they will be analyzed as covariates, including subgroup and meta-regression analyses (lines 105-107; lines 257-260; 267-274)

Clarify for outcomes such as discharge, mortality, and complications how you will ensure that these outcomes are attributable to malaria (e.g. restriction to lab-confirmed clinical malaria, handling of co-morbidities and mixed-cause deaths) and define any time frames used.

Authors’ response

The details of outcomes have been included as a sub-section on clinical outcomes in the methods and further details provided ion attribution of the outcomes to malaria or otherwise ncluding how co-morbidities will be handled, causes of death and time frame (226-242)

Justify the choice of excluding studies with fewer than 100 participants and, if feasible, pre-specify a sensitivity analysis, including these smaller studies, to assess their impact.

Authors’ response

We anticipated that high heterogeneity may be impacted by sample size, and most observational studies will be cross-sectional studies. This, along with other reasons for the proposed exclusion, has been added to the main text of the manuscript. As suggested, we intend to re-include studies with fewer than 100 participants during the sensitivity analysis and observe their impact (lines 128-133;281-283).

Make explicit in the Methods that full search strings for all databases are provided in a supplementary file (e.g. "Search string for the study-malaria") and ensure that the example PubMed strategy in the text matches the supplementary material.

Authors’ response

Thank you for the observation-the prototype PubMed search string has been included in text and full search strategy for all data bases are added as supplementary files (lines 155-177).

Please strengthen the Data Synthesis and Statistical Analysis section by adding appropriate methodological references for the random-effects model, the use of R ("meta" and "metafor" packages), and key measures such as I² and prediction intervals.

Authors’ response

Thank you for your observation. The data synthesis and statistical analysis sections have been updated with appropriate references ((lines 246, 251. 254. 255)

Add supporting methodological references for the Assessment of Publication Bias and Sensitivity Analysis (e.g. Egger’s test, funnel plots, trim-and-fill, leave-one-out analyses) and briefly clarify how results from these procedures will be interpreted.

Authors’ response

This section of the data synthesis and statistical analysis sections has also been updated with appropriate references. In addition, further details of their interpretations have been added, as suggested (lines 275-288)

Provide a more detailed, prespecified plan for subgroup analyses and any meta-regression (e.g. listing subgroup variables, minimum number of studies per subgroup/covariate, and the basic modelling approach).

Authors’ response

The details of the proposed subgroup analyses, including the minimum number of studies to be considered for the subgroup/covariate, have been added, including the approach to modelling/meta-regression, in the manuscript (lines 290-304).

Consider adding an explicit "Strengths and limitations" subsection that also summarizes the main anticipated limitations of the planned review (e.g. heterogeneity of definitions, risk of publication/language bias, hospital-based data).

Authors’ response

A subsection on the strengths and limitations has been added, highlighting the suggested observations and points raised (lines 340 -357).

Please revise the manuscript carefully for formatting, structure, and English language clarity (e.g. consistent terminology, numbering, headings) so that it meets PLOS

ONE's readability and presentation standards.

Authors’ response

Thanks for the observation-The manuscript has been reformatted in keeping with PLOS ONE style and to enhance readability

6. Review Comments to the Author

You may also provide optional suggestions and comments to authors that they might find helpful in planning their study.

Reviewer #1: I noticed a couple of inconsistencies that should be clarified. For example, the abstract says only observational studies will be eligible, but the eligibility criteria section later mentions both observational and interventional studies. Similarly, the abstract limits inclusion to studies published in English, while the eligibility criteria section states that studies in any language will be included.

Authors’ response

Thank you for your observations. The mix-up has been corrected. The abstract and main texts have been harmonized with regard to the exclusion criteria and study language. The search is proposed to be without language restriction (lines 34-37)

Some points also need more detail. The cutoff of 100 participants for exclusion is mentioned, but the rationale for this threshold isn’t explained.

Authors’ response

The reasons for the proposed exclusion of studies with fewer than 100 participants were included (lines 128-133).

The secondary outcomes were also vague. For instance, “associated factors” isn’t clearly defined, and for outcomes like discharge, mortality, and complications, it’s not clear how the authors will ensure these outcomes are directly attributable to malaria.

Authors’ response

Thank you for the clarification. We have provided the details of what associated factors entail and further provided information on covariate analysis, including meta-regression analyses. (Lines 105-107;290-304)

In the exclusion criteria, “asymptomatic malaria” should be explicitly defined to avoid ambiguity.

Authors’ response

The clear definition of asymptomatic malaria has been included. (lines 209-210)

For the search strategy, I’d suggest adding a supplementary file that lists all search strings used across databases. That would make the process more transparent and reproducible for readers.

Authors’ response

A supplementary file of full the search string across the databases have been as well as supplementary file as suggested (supplementary file 2)

In the methods section, some areas like Data Synthesis and Statistical Analysis and Assessment of Publication Bias and Sensitivity Analysis don’t have any supporting references. Even if these are common methods, citing references would strengthen the protocol.

Authors’ response

References have been added as appropriate for the data synthesis and statistical analysis part as suggested (lines 275-288)

I also don’t see where the subgroup analysis methodology is clearly and detailedly stated.

Authors’ response

The details of the subgroup analysis have been included, as suggested. (Lines 290-304)

An “Expected Limitations” section would help.

Authors’ response

We have included expectations limitations of the study. (Lines 350-357)

Finally, there are many formatting and arrangement issues throughout the manuscript that don’t meet standards, which may lead to difficulty in reading for readers.

Authors’ response

The formatting has been revised to be in consonant with PLOS ONE’s style and enhance readability

Reviewer #2: Dear Authors,

Well done for the effort in drafting this protocol.

Authors’ response

We appreciated this positive feedback with thanks

Please address the concerns below.

Abstract: Line 26-28

Clarify how your study design will fill the gap (i.e. unknown exact burden).

Authors’ response

This background statement has been revised to ensure that it capture how the study design will answer the research gap (unknown exact burden). (Lines 26-30; 83-87)

Methods:

Line 32 reads "Eligible studies will be observational and published in English from inception till June 30, 2025.

Authors’ response

Lines 105-107 also reads "Study design (S): This review will include quantitative observational and interventional studies, including cross-sectional, cohort, case-control, and longitudinal studies, conducted in Nigeria and published in any language."

Authors’ response

The mix up in this statement has been revised and harmonized. The study will be conducted without language restrictions. It will also include both observation and interventional study provided they meet the study inclusion criteria

First statement has observational studies (no indication of interventional studies) and language restriction. The second statement has interventional studies and no language

restriction. Reconcile the two statements and include the necessary justifications.

Authors’ response

The mix up in this statement has been revised and harmonized. The study will be conducted without language restrictions. It will also include both observation and interventional study provided they meet the study inclusion criteria. (Lines 34-37)

Exclusion criteria:

What happens to studies involving adult malaria in your systematic search?

Authors’ response

Studies that do not include discernible pediatric data with a clear denominator will be excluded.

Line 116 reads "4. Studies with a sample size of fewer than 100 participants."

Justify clearly (with reference, if any) why you will exclude studies with specifically less than 100 sample size.

Authors’ response

The rationale for the exclusion of studies with less than 100 sample size have been included. (Lines 128-133)

Line 119 reads "6. Studies on asymptomatic malaria."

What do you mean by asymptomatic malaria? Justify its exclusion.

Authors’ response

Definition of asymptomatic malaria have been included including the rationale for their exclusion. (Lines 136-141)

Provide clarity in the two sentences below.

144 ... "Level 1 would involve screening of identified studies for the

145 study design quantitative observational and interventional studies would be accepted."

Authors’ response

These two statement have been re-written to improve their clarity (lines 184-186)

151 ... "The screened will be done

152 independently by two reviewers..."

Authors’ response

These statements have been re-written to improve their clarity (lines 191-193)

In lines 176-177, citation is required for the statement "The NOS tool evaluates observational studies based on three key domains: selection of study groups, comparability, and ascertainment of outcomes or exposures."

Authors’ response

A citation has been added as suggested.

All statements under authors’ contributions (lines 271-278) appear to suggest that search, data extraction, and analysis have already been carried on. Please rectify.

Authors’ response

Thank you for your observation. While a few actions have been carried out (conceptualization and study design). The search and subsequent processes have yet to begin. This section has been re-written to indicate future plan actions. (Lines 393-402)

Reviewer #3: Thanks for your impactful study. Comments:

The abstract initially states inclusion of English-language studies only, while later sections state “any language” with translation plans. Please ensure full consistency across the abstract, methods, eligibility criteria, and search strategy. The revised “no language restriction” approach is methodologically appropriate, but it must be reflected uniformly throughout the manuscript.

Authors’ response

Thank you for your observations. We have revised the manuscript and harmonized the inclusion criteria and the language. The studies will include both observational studies and interventional studies provided they meet the inclusion criteria, while the language will be without restrictions. (Lines 34-37).

Given the wide variation expected across Nigerian geopolitical zones, diagnostic methods, healthcare levels, and time periods, heterogeneity is likely to be substantial. Please predefine thresholds for when meta-analysis will not be performed due to excessive heterogeneity and clarify how such situations will be handled.

Authors’ response

Many thanks for the observation-we have added predefined thresholds for pooled data to enhance the meta-analysis findings. (Lines 290-304)

The protocol states that odds ratios will be pooled for associated factors. However, observational studies may report heterogeneous effect m

---

## [Decision Letter · Decision Letter 1]

2 Mar 2026

Burden, risk factors, and clinical outcomes of pediatric malaria in Nigeria: A systematic review and meta-analysis protocol

PONE-D-25-56196R1

Dear Dr. Olayinka,

We’re pleased to inform you that your manuscript has been judged scientifically suitable for publication and will be formally accepted for publication once it meets all outstanding technical requirements.

Kind regards,

Taiwo Opeyemi Aremu, MD, MPH, PhD

Academic Editor

PLOS One

Additional Editor Comments (optional):

Reviewers' comments:

Reviewer's Responses to Questions

**Comments to the Author**

1. Does the manuscript provide a valid rationale for the proposed study, with clearly identified and justified research questions?

Reviewer #2: Yes

Reviewer #3: Yes

2. Is the protocol technically sound and planned in a manner that will lead to a meaningful outcome and allow testing the stated hypotheses?

Reviewer #2: Yes

Reviewer #3: Yes

3. Is the methodology feasible and described in sufficient detail to allow the work to be replicable?

Reviewer #2: Yes

Reviewer #3: Yes

4. Have the authors described where all data underlying the findings will be made available when the study is complete?

Reviewer #2: Yes

Reviewer #3: Yes

5. Is the manuscript presented in an intelligible fashion and written in standard English?

Reviewer #2: Yes

Reviewer #3: Yes

You may also provide optional suggestions and comments to authors that they might find helpful in planning their study.

Reviewer #2: I am satisfied with the responses and revised version. Methodological inconsistencies in the original submission have been addressed by the authors.

Reviewer #3: Thanks for revised version and answer the concerns. This protocol addresses an important gap by aiming to synthesize national estimates of pediatric malaria burden, associated factors, and outcomes in Nigeria. The methodology is generally sound, with predefined subgroup and meta-regression analyses and clear outcome attribution; however, some issues require clarification. The concept of “burden” should be more precisely defined (prevalence vs. broader measures such as hospitalization and mortality), and the rationale for including interventional studies in prevalence pooling needs stronger justification. Overall, the study is relevant and well structured.

**Do you want your identity to be public for this peer review?** For information about this choice, including consent withdrawal, please see our Privacy Policy

Reviewer #2: No

Reviewer #3: No

---

## [Editor Report · Acceptance letter]

PONE-D-25-56196R1

PLOS One

Dear Dr. Olayinka,

I'm pleased to inform you that your manuscript has been deemed suitable for publication in PLOS One. Congratulations! Your manuscript is now being handed over to our production team.

Kind regards,

on behalf of

Dr. Taiwo Opeyemi Aremu

Academic Editor

PLOS One